

# A risk score model with five long non-coding RNAs for predicting prognosis in gastric cancer: an integrated analysis combining TCGA and GEO datasets

Yiguo Wu[1,*], Junping Deng[2,*], Shuhui Lai[1], Yujuan You[3] and Jing Wu[4]

[1] Department of Medicine, Nanchang University, Nan Chang, China
[2] Department of General Surgery, The First Affiliated Hospital of Nanchang University, Nan Chang, China
[3] Department of Anesthesiology, The Second Affiliated Hospital of Nanchang University, Nan Chang, China
[4] Shenzhen Prevention and Treatment Center for Occupational Diseases, Shen Zhen, China
[*] These authors contributed equally to this work.

Corresponding authors
Yujuan You, 506737972@qq.com
Jing Wu, 446346807@qq.com

## ABSTRACT

**Background**. Gastric cancer (GC) is one of the most common carcinomas of the digestive tract, and the prognosis for these patients may be poor. There is evidence that some long non-coding RNAs(lncRNAs) can predict the prognosis of patients with GC. However, few lncRNA signatures have been used to predict prognosis. Herein, we aimed to construct a risk score model based on the expression of five lncRNAs to predict the prognosis of patients with GC and provide new potential therapeutic targets.
**Methods**. We performed differentially expressed and survival analyses to identify differentially expressed survival-ralated lncRNAs by using GC patient expression profile data from The Cancer Genome Atlas (TCGA) database. We then established a formula including five lncRNAs to predict the prognosis of patients with GC. In addition, to verify the prognostic value of this risk score model, two independent Gene Expression Omnibus (GEO) datasets, GSE62254 ($N = 300$) and GSE15459 ($N = 200$), were employed as validation groups.
**Results**. Based on the characteristics of five lncRNAs, patients with GC were divided into high or low risk subgroups. The prognostic value of the risk score model with five lncRNAs was confirmed in both TCGA and the two independent GEO datasets. Furthermore, stratification analysis results showed that this model had an independent prognostic value in patients with stage II–IV GC. We constructed a nomogram model combining clinical factors and the five lncRNAs to increase the accuracy of prognostic prediction. Enrichment analysis based on the Kyoto Encyclopedia of Genes and Genomes (KEGG) suggested that the five lncRNAs are associated with multiple cancer occurrence and progression-related pathways.
**Conclusion**. The risk score model including five lncRNAs can predict the prognosis of patients with GC, especially those with stage II-IV, and may provide potential therapeutic targets in future.

## INTRODUCTION

Gastric cancer (GC) is one of the most common carcinomas of the gastrointestinal (GI) tract and is particularly prevalent in Asian countries. It is estimated that approximately 679,100 individuals were diagnosed with GC in 2015 in China and approximately 498,000 of them died that same year (*Saka et al., 2011*; *Chen et al., 2016*). The standard therapies for GC are surgery and chemotherapy. However, most patients with advanced GC show recurrence of the malignancy and metastasis after treatment, resulting in poor prognosis. Despite considerable research in therapies for GC, the prospects of survival of patients with GC remain bleak (*Saka et al., 2011*). The identification of patients with GC with poor prognosis and the administration of effective treatment as early as possible are key to improving survival. The investigation of potential therapeutic and prognostic biomarkers for GC is of considerable importance.

Long non-coding RNAs (lncRNAs) are RNAs with lengths of $\geq 200$ nucleotides with no or limited protein-coding potential. There is considerable evidence that lncRNAs play crucial roles in the initiation and developments of cancers. For example, lncRNA-ATB disorders contribute to cancer cell proliferation, migration, invasion, and drug-resistance as well as induce epithelial-mesenchymal transition by competitively binding to microRNAs (*Li et al., 2017*; *Balas & Johnson, 2018*). Some researchers have suggested that lncRNAs serve as new prognostic biomarkers in various cancers, including CCAT2 (*Yu et al., 2017*), HOXB-AS3 (*Huang et al., 2017*), and ASLNC07322 (*Li et al., 2019*) in colon cancer. Many lncRNAs closely related to the prognosis of patients with GC have been identified, including MEG3 (*Wei & Wang, 2017*), SNHG7 (*Wang et al., 2017*), and DANCR (*Mao et al., 2017*). Risk score models have also been constructed to predict the prognosis of human cancers. The differences in prognosis in non–small-cell lung cancer can be identified by its eight-lncRNA signature (*Miao et al., 2019*). However, the identification of lncRNAs related to the prognosis of patients with GC is still in its early stages and additional research is warranted.

In this study, we analyzed the data of 450 patients with GC from The Cancer Genome Atlas (TCGA) database to identify differentially expressed lncRNAs for the prognostic prediction. We used two independent Gene Expression Omnibus (GEO) (*Barrett et al., 2013*) datasets to validate the selected-lncRNAs. In addition, we analyzed the accuracy of the prediction of five lncRNAs in different clinical subgroups using lncRNA data in combination with the clinical characteristics of the patients. Furthermore, we constructed a nomogram model by combining clinical factors and five lncRNAs to increase the accuracy of prognostic prediction. Finally, we performed pathway enrichment analysis to determine the potential functions of these lncRNAs in GC.

## MATERIALS & METHODS

### Preparation of GC datasets

We acquired a training dataset of GC samples from TCGA at UCSC Xena (https: //xenabrowser.net/) before February 1st, 2019 comprising 450 samples and 14147 lncRNAs (case: normal = 414:36). We used these 450 samples to perform differential expression

analysis. After excluding six cases with missing overall survival (OS) prognostic information, 408 cases were included for further univariate Cox proportional hazard regression analysis and subsequent analysis in the training group. The microarray data for the validation group and survival data of the patients are publicly available at GEO with accession numbers GSE62254 ($N = 300$; 1397 lncRNAs) and GSE15459 ($N = 200$; 1397 lncRNAs).

## Normalization of GEO data

Because the two GEO datasets (GSE62254 and GSE15459) had different expression profiles, we performed quantile normalization on the original data and downloaded it as a probe-level CEL file. Affymetrix U133 Plus 2.0 was used as the probe matching platform. Data were downloaded from the Affymetrix website (http://www.affymetrix.com), and a total of 2986 lncRNA-specific probes were included.

## Construction of an lncRNA-based risk score model from the training group

The lncRNAs that were differentially expressed between GC and normal gastric tissue in TCGA dataset were identified using the "limma" R package of the R statistical computing environment (log2|fold change| >1 and adjusted $P < 0.05$), and the adjusted $P$ value was used to reduce false positives (*Deng, Xu & Wang, 2019*; *Zeng et al., 2019*). The candidate lncRNAs were analyzed using univariate Cox proportional hazard regression analysis ($P < 0.05$). The cutoff values of lncRNA expression were determined as the median of all expression values in Cox survival analysis. In total, we identified 278 lncRNAs with statistically significant differences. After identifying the lncRNAs common to both TCGA and GEO (GSE62254) datasets, we performed multivariate Cox proportional hazards analysis to identify independent prognostic lncRNAs. Finally, we constructed an lncRNA-based risk score model from a linear combination of the expression levels of these lncRNAs, multiplied by the regression coefficients obtained from the multivariate Cox proportional hazards regression analysis.

## Validation of the lncRNA-based model for prognostic prediction

We calculated the risk scores of each patient and used the corresponding median score as the cutoff value to classify them into two groups: high risk and low risk subgroups. We used Kaplan–Meier analysis to compare the survival of the two groups and time-dependent receiver operating characteristic (ROC) curves to assess our lncRNA-based risk model. We used two GEO datasets to validate the model for prognostic prediction. Cox proportional hazards regression analysis was used to estimate the hazard ratio (HR) of the model with 95% confidence intervals to further evaluate the predictive value of the model for each clinical subgroup. Clinical subgroups were determined based on sex, tumor–node–metastases (TNM) stage, histologic grade, race, and age. Finally, we constructed a nomogram combining the model with clinical factors using the "rms" package. We also calculated the concordance index (C-index) and plotted a calibration curve to determine its predictive accuracy and discriminatory capacity.

## Potential functions of the five lncRNAs

To determine the potential functions of the five lncRNAs, which appeared to be discriminatory, we performed linear regression analysis of the relationship between the lncRNAs and the protein-coding genes in TCGA dataset. The screening criteria for the protein-coding genes were a positive association with at least one lncRNA (Pearson coefficient > 0.4). After identifying the candidate genes, we screened out aberrantly activated signaling pathways using the Kyoto Encyclopedia of Genes and Genomes (KEGG) enrichment analysis via web-based Gene Set Analysis Toolkit (http://www.webgestalt.org/), a popular software tool for functional enrichment analysis related to KEGG pathways (*Yang et al., 2019*; *Wang et al., 2013*).

## Statistical analysis

We used the R software (version 3.6.1) for statistical analyses. Differentially expressed analysis was performed using the "limma" R package. Univariate and multivariate Cox proportional hazards regression analyses were performed to identify prognosis-related lncRNAs. The "survival" and "survminer" packages were used for Cox proportional hazards regression analyses, Kaplan–Meier survival analysis and calculation of C-index. A time-dependent ROC curve to assess the specificity and sensitivity of the risk score model was constructed using the "survivalROC" package. A nomogram combining the risk score model with the clinical factors was constructed using the "rms" package. The Review Manager software (version 5.3) was used to construct a forest plot. Chi-square tests were used to compare the recurrence and mortality rates between the high and low risk subgroups. A $P$-value of <0.05 was considered statistically significant, and all tests were two sided. Pearson's linear regression analysis was used to determine the relationship between lncRNAs and protein-coding genes.

# RESULTS

## Identification of five prognostic lncRNAs

The datasets are publicly available and recruitment has already happened. We performed differentially expressed analysis (log2|fold change| >1 and adjusted $P < 0.05$) and univariate Cox proportional hazard regression analysis ($P < 0.05$) to identify survival-related lncRNAs. A total of 278 lncRNAs were analyzed further. To validate the predictive accuracy, we compared the lncRNAs selected from TCGA database with the GEO validation group. We found that 37 lncRNAs were common between the 278 lncRNAs and the validation dataset (GSE62254) (Table S1). Multivariate Cox proportional hazards regression analyses identified five lncRNAs as independent prognostic factors of GC: LINC00205, TRHDE-AS1, OVAAL, LINC00106, and MIR100HG (Table 1). Figs. 1A–1B shows the expression profiles of the five lncRNAs in patients with GC as volcano and heat maps, and Figs. 1C–1D shows the survival curves based on the OS and disease-free survival (DFS) of the 408 patients. Owing to the lack of clinical data in GSE15459, Table 2 shows the clinical features of patients with GC in the training group and GSE62254.

**Table 1  Five lncRNAs significantly associated with prognosis of GC patients in the training group.** Derived from the multivariable Cox proportional hazards regression analysis in the training group.

| LncRNA name | Ensemble ID | Chr. | Coordinate | Coefficient | Hazard ratio | P value |
|---|---|---|---|---|---|---|
| LINC00205 | ENSG00000223768.1 | 21 | 45288052–45297354 | 0.249092 | 1.373451497 | 0.047216345 |
| TRHDE-AS1 | ENSG00000236333.3 | 12 | 72253507–72273509 | 0.182045 | 1.846654514 | 0.000109193 |
| OVAAL | ENSG00000236719.2 | 1 | 180558974–180566518 | 0.271169 | 1.880897277 | 0.0000744 |
| LINC00106 | ENSG00000236871.6 | X&Y | 1397025–1399412 | −0.207942 | 0.624972486 | 0.003469142 |
| MIR100HG | ENSG00000255248.6 | 11 | 122028329-122422871 | 0.502539 | 1.396343319 | 0.036829012 |

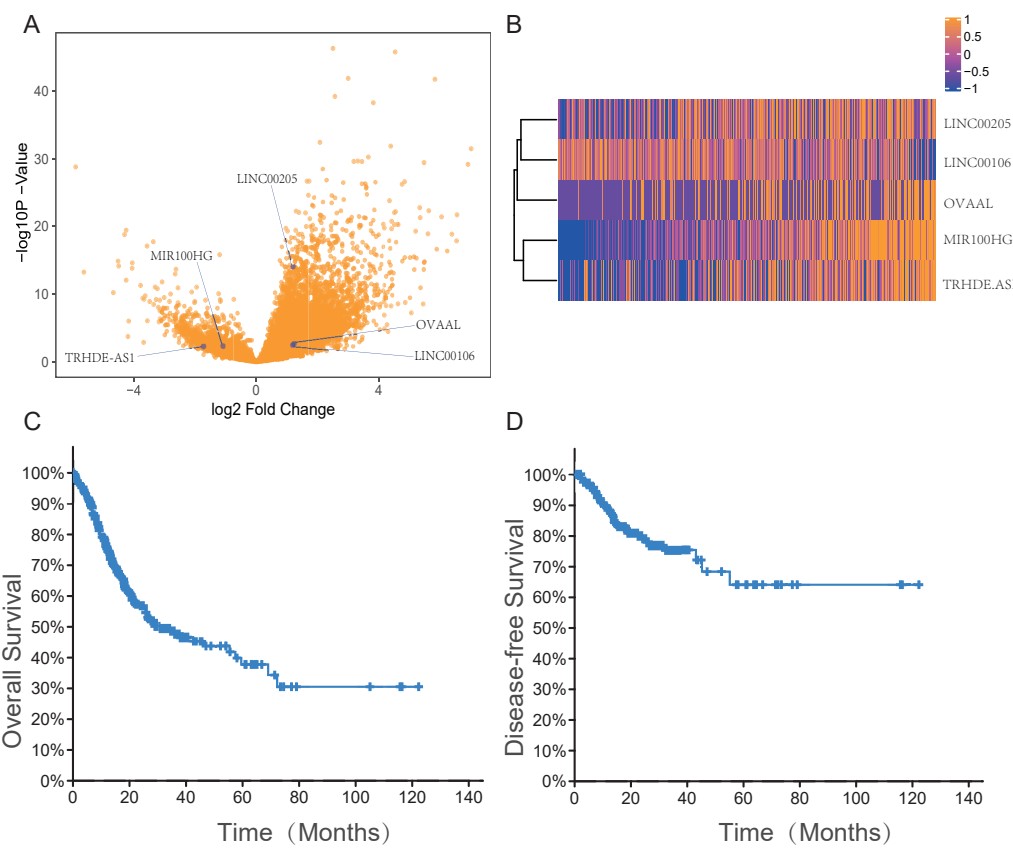

**Figure 1  The expression information of five lncRNAs, overall survival and disease free survival in gastric cancer patients in the TCGA dataset.** (A) Volcano plot with blue dots indicating five lncRNAs expression levels which is significantly different between tumor and normal tissue based on the criteria of an absolute log2 fold change (FC) >1 and adjusted $P < 0.05$. (B) Heatmap of the five-lncRNA expression profile of the 414 patients in the TCGA dataset. Among five lncRNAs, MIR100HG and TRHDE-AS1 have a similar expression in 414 patients in the TCGA dataset, otherwise the other three lncRNAs do as well. (C–D) The survival curves based on the OS and DFS of the 408 patients in TCGA dataset.

## Construction of an lncRNA-based risk model from the training group

According to the schematic workflow of this study (Table 3), using the coefficients of the five lncRNAs identified by multivariate Cox hazard analysis, we created a risk-score formula as follows: risk score = (0.249092 × expression level of LINC00205) + (0.182045 ×

**Table 2 The clinical features of GC patients in training group and GSE62254.**

| | Training group | | Validation group-1 (GSE62254) | |
|---|---|---|---|---|
| **Variables** | **n = 408** | **%** | **n = 300** | **%** |
| **Gender** | | | | |
| Male | 263 | 64.46 | 199 | 66.33 |
| Female | 145 | 35.54 | 101 | 33.67 |
| **Age** | | | | |
| Old (≥50 years old) | 377 | 92.40 | 262 | 87.33 |
| Young (<50 years old) | 31 | 7.60 | 38 | 12.67 |
| **TNM stage** | | | | |
| Stage I | 55 | 13.48 | 30 | 10.00 |
| Stage II | 120 | 29.41 | 96 | 32.00 |
| Stage III | 167 | 40.93 | 95 | 31.67 |
| Stage IV | 41 | 10.05 | 79 | 26.33 |
| Not Available | 25 | 6.13 | 0 | |
| **T stage** | | | | |
| T1 | 20 | 4.90 | 2 | 0.67 |
| T2 | 87 | 21.32 | 186 | 62.00 |
| T3 | 178 | 43.63 | 91 | 30.33 |
| T4 | 114 | 27.94 | 21 | 7.00 |
| TX | 9 | 2.21 | 0 | |
| **N stage** | | | | |
| N0 | 120 | 29.41 | 38 | 12.67 |
| N1 | 110 | 26.96 | 131 | 43.67 |
| N2 | 77 | 18.87 | 80 | 26.67 |
| N3 | 82 | 20.10 | 51 | 17.00 |
| NX | 17 | 4.17 | 0 | |
| Not Available | 2 | 0.49 | 0 | |
| **M stage** | | | | |
| M0 | 362 | 88.73 | 273 | 91.00 |
| M1 | 27 | 6.62 | 27 | 9.00 |
| MX | 19 | 4.66 | 0 | |
| **Survival status** | | | | |
| Alive | 251 | 61.52 | 148 | 49.33 |
| Dead | 157 | 38.48 | 152 | 50.67 |

expression level of TRHDE-AS1) + (0.271169 × expression level of OVAAL) + (−0.20794 × expression level of LINC00106) + (0.502539 × expression level of MIR100HG). Among the five lncRNAs, LINC00106, which had a negative coefficient, was considered as a protective factor. The remaining four lncRNAs with positive coefficients, namely LINC00205, TRHDE-AS1, OVAAL, and MIR100HG, were risk factors. The risk scores of each patient in the training group were calculated (Table S2), ranging from −2.086959745 to 2.270305234. The patients in the training group were divided into two subgroups: high risk (n = 204) and low risk (n = 204) subgroups, with the median score (−0.001085) as the cutoff value. We performed Kaplan–Meier survival analysis to assess the effect of the

**Table 3** The schematic workflow of the present study.

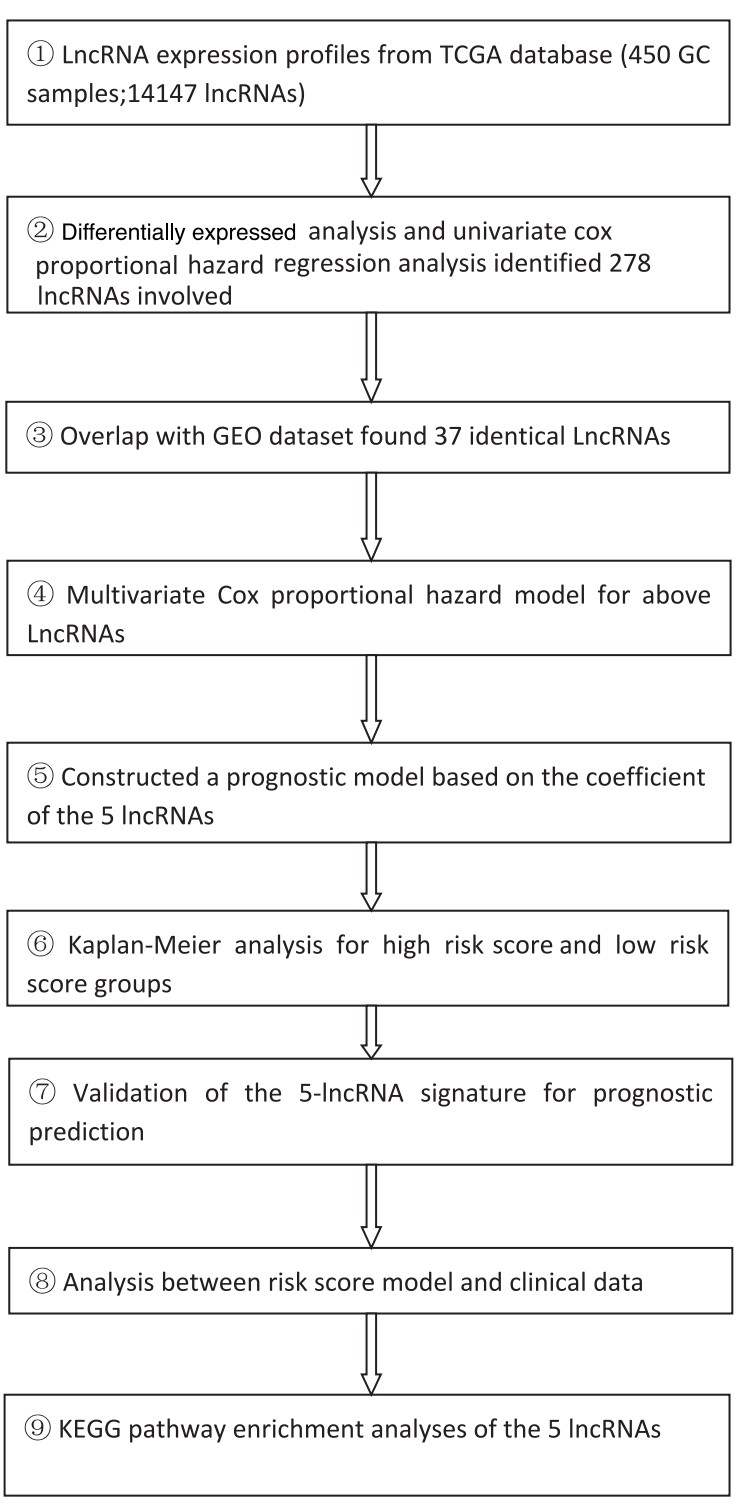

lncRNA-based risk model on the OS and DFS of patients with GC in the training group (Figs. 2A–2B). Our results indicated that the high-risk group had a significantly poorer prognosis than the low risk group for both OS and DFS ($P = 1 \times 10^{-6}$ and $6 \times 10^{-6}$, respectively). Figsures 2C–2F shows the scatter plots of the recurrence and mortality rates of patients with GC. The recurrence and mortality rates were significantly higher in the high risk group than in the low risk group ($P < 0.001$). To accurately evaluate the prognostic value of the five-lncRNA signature, we performed time-dependent ROC analysis using the 1–4 years cutoff value of OS and the 1–2 years cutoff of DFS as the ROC ending points (Figs. 2G–2H and Figs. S1A–S1D). The area under the ROC curve (AUC) was 0.734 for the 4-year cutoff value of OS and 0.692 for the 2-year cutoff value of DFS, respectively, and had the highest predictive value among those years, indicating that our model can be used for survival prediction in patients with GC (Figs. 2G–2H).

## Validation of the lncRNA-based risk score model for prognostic prediction in independent groups

To assess the prognostic significance of this novel lncRNA-based risk model including the five-lncRNA signature in patients with GC, we used the other two independent validation datasets from the GEO database. We calculated the risk score using the formula mentioned above (Table S2). The patients with GC in GSE62254 (validation group-1; $N = 300$) and GSE15459 (validation group-2; $N = 200$) datasets were divided into high risk and low risk groups according to the median risk score. Owing to the lack of DFS data in GSE15459, we only calculated the OS of the patients. The high risk group had a poorer OS than the low risk group (log-rank $P = 0.01$) (Figs. 3A–3B). Figures 3C–3D shows the scatter plots for the death events. The mortality rates were significantly higher in the high risk group than in the low risk group ($P < 0.001$). The AUC for the two validation groups in the 4-year cutoff OS was 0.622 and 0.610 for validation group 1 and 2, respectively (Figs. 3E–3F). Figures S2A–S2F shows the ROC curve for the 1–3 year cutoff OS for the validation groups 1 and 2. Furthermore, we verified the performance of our risk score model for DFS of the GSE62254 dataset (Figs. S3A–Fig. S3D). Our results further confirmed the value and robustness of this risk score model for prognostic prediction in patients with GC.

## The lncRNA-based risk model has a favorable prognostic prediction in patients with stage II, III, and IV

To further investigate the performance of our lncRNA-based risk model, we performed stratified Kaplan–Meier survival analysis of OS in the training group based on the AJCC TNM stages I, II, III, or IV (Figs. 4A–4D). The five-lncRNA signature showed good predictive value for OS of stages II ($P = 0.008$), III ($P = 0.02$), and IV ($P = 0.01$), but not I ($P = 0.3$).

To estimate the HR of each subgroup of patients as defined by sex, TNM stage, histologic grade, race and age($\geq$ or $< 50$ years) (Table 4), we used our model to divide the patients into two risk groups on the basis of the median cutoff value. Forest plots are shown in Fig. 5. Table S3 shows the HR of each subgroup of patients in GSE62252. The risk score model had a relatively good prognostic value in the clinical subgroups of sex, histologic grade and age. To improve the prognostic value of this model, we combined the clinical factors

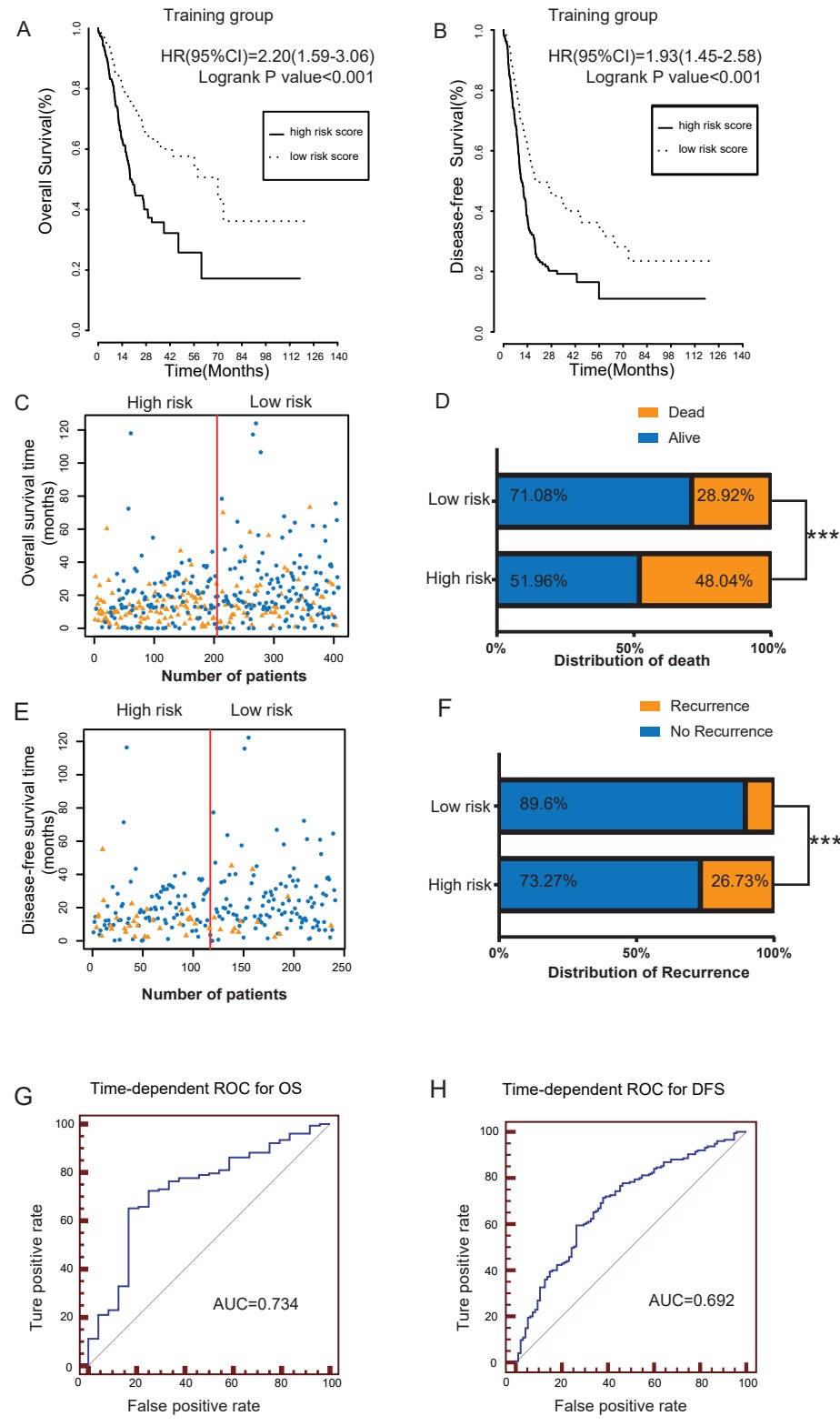

**Figure 2** **The prognostic value of lncRNA-based risk model in training group.** (A–B) Kaplan–Meier analysis of patients' OS and DFS in the high risk (n = 204) and 

**Figure 2 (…continued)**
low risk ($n = 204$) subgroups of the training group. (C) The scatter plot of lncRNA-based risk model distribution for patient survival status. (D) The percentage of patient survival status in the high risk and low risk subgroups of the training group. (E) The lncRNA-based risk model distribution for patient recurrence. (F) The percentage of patient recurrence in the high risk and low risk subgroups of the training group. (G–H) The time-dependent ROC analysis of the risk score for prediction the 4-year cutoff OS and 2-year cutoff DFS of the training group. The area under the curve was calculated for ROC curve. *** $P <$ 0.001.

with the risk score model to construct a nomogram model for prognostic prediction. The nomogram model and calibration curve are shown in Figs. 6A–6B. To evaluate the effect of the nomogram model, we calculated its C-index. The C-index for predicting the 4-year OS of patients with GC was 0.69668, indicating that this model is a valuable indicator for prognostic prediction.

### Potential functions of the five lncRNAs

To investigate the functions of the five lncRNAs in patients with GC, we calculated Pearson correlations between the five-lncRNA signature and 19,605 protein-coding genes in TCGA dataset. A total of 3069 genes (Table S4) were positively correlated with at least one lncRNA (Pearson's coefficient $> 0.4$) (Fig. 7A). We further selected these genes for KEGG pathway enrichment analysis. By ranking based on $-\log P$ value ($Q$ value), we selected the top 10 pathways for construction of a bubble plot (Fig. 7B) (*Zeng et al., 2019*; *Deng, Xu & Wang, 2019*). For biological processes, the co-expressed genes were mainly enriched in pathways involved in cancer, such as the focal adhesion pathway, cGMP$-$PKG signaling pathway and calcium signaling pathway. This finding indicates that the five lncRNAs may be involved in the regulation of tumor initiation and progression.

## DISCUSSION

In this study, we identified a potential signature involving five lncRNAs that are differentially expressed in tumor and normal tissues, and might be valuable for prognostic prediction in GC. The prognostic performance of this lncRNA-based risk score model was verified using both TCGA and GEO datasets. Stratified analysis suggested that the risk score model is valuable for prognostic prediction in patients with stage II-IV GC. To enhance the predictive accuracy of the model, we combined clinical parameters with the five-lncRNA signature to construct a nomogram model and confirmed its performance using a calibration curve and C index.

GC is a common malignancy of the GI tract (*Siegel, Miller & Jemal, 2019*). Despite continuous improvements in treatment, the 5-year survival rate of patients with advanced GC is only approximately 20% (*Min et al., 2019*; *Misawa et al., 2019*). Therefore, early diagnosis, early identification of high-risk patients and implementation of effective treatment measures as early as possible are necessary to improve survival. It is also important to develop novel prognostic indicators of GC. Over the past few decades, research has shown that protein-coding genes(*Ghoorun et al., 2019*; *Luo et al., 2019*) and microRNAs (*Li et al., 2020*; *Zhou, Wu & Bi, 2019*), play vital roles in the occurrence and development of various

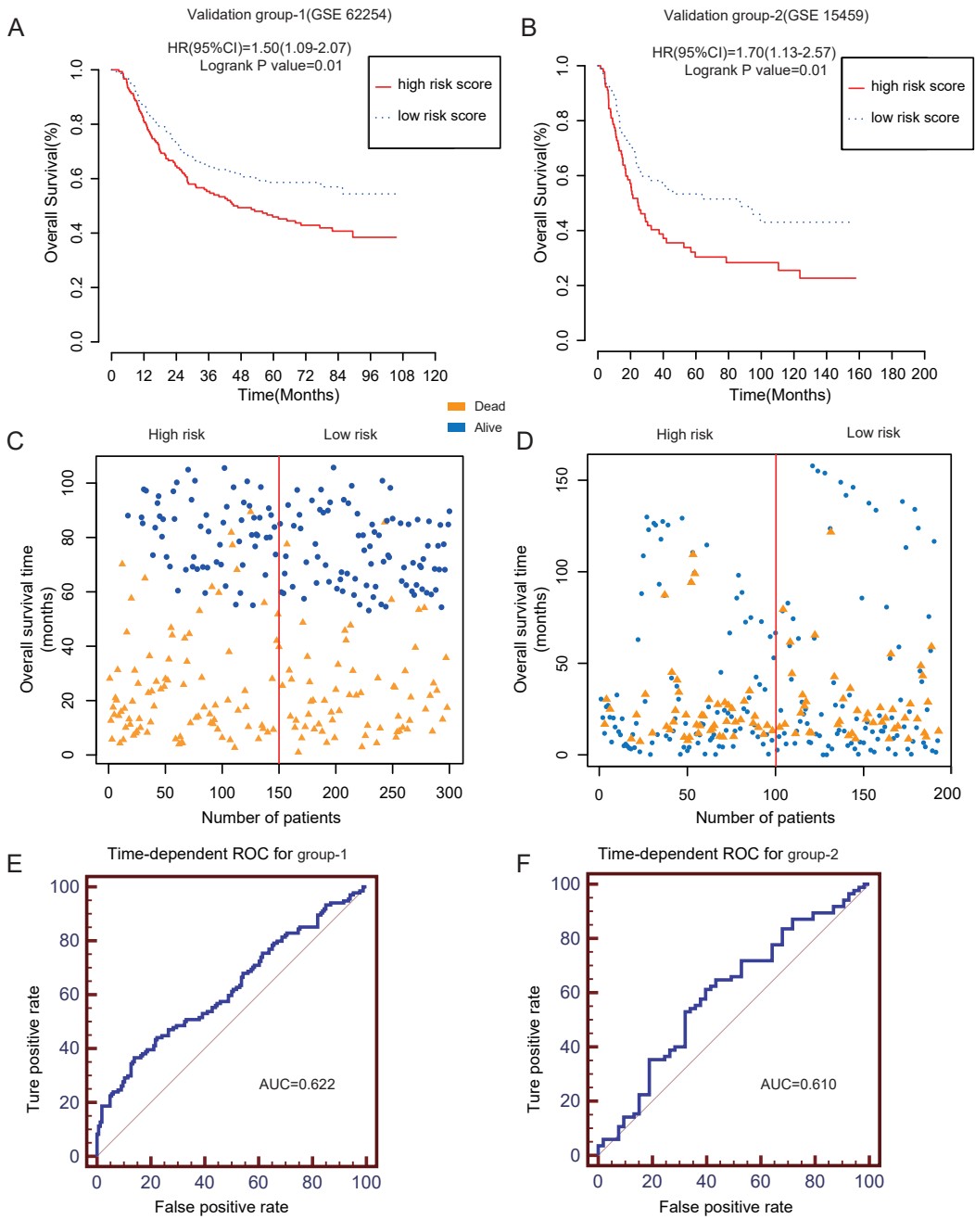

**Figure 3** **The prognostic value of lncRNA-based risk model in two independent GEO validation groups.** (A–B) Kaplan–Meier analysis of predicting OS of GC patients based on the high risk and low risk subgroups in two independent validation groups (GSE62254 and GSE15459). (C–D) The scatter plot of five-lncRNA-based risk score distribution for patient survival status in two independent validation groups.(E–F) The time-dependent ROC analysis of the risk score for prediction the 4-year cutoff OS of the two independent validation groups. The area under the curve was calculated for ROC curve.

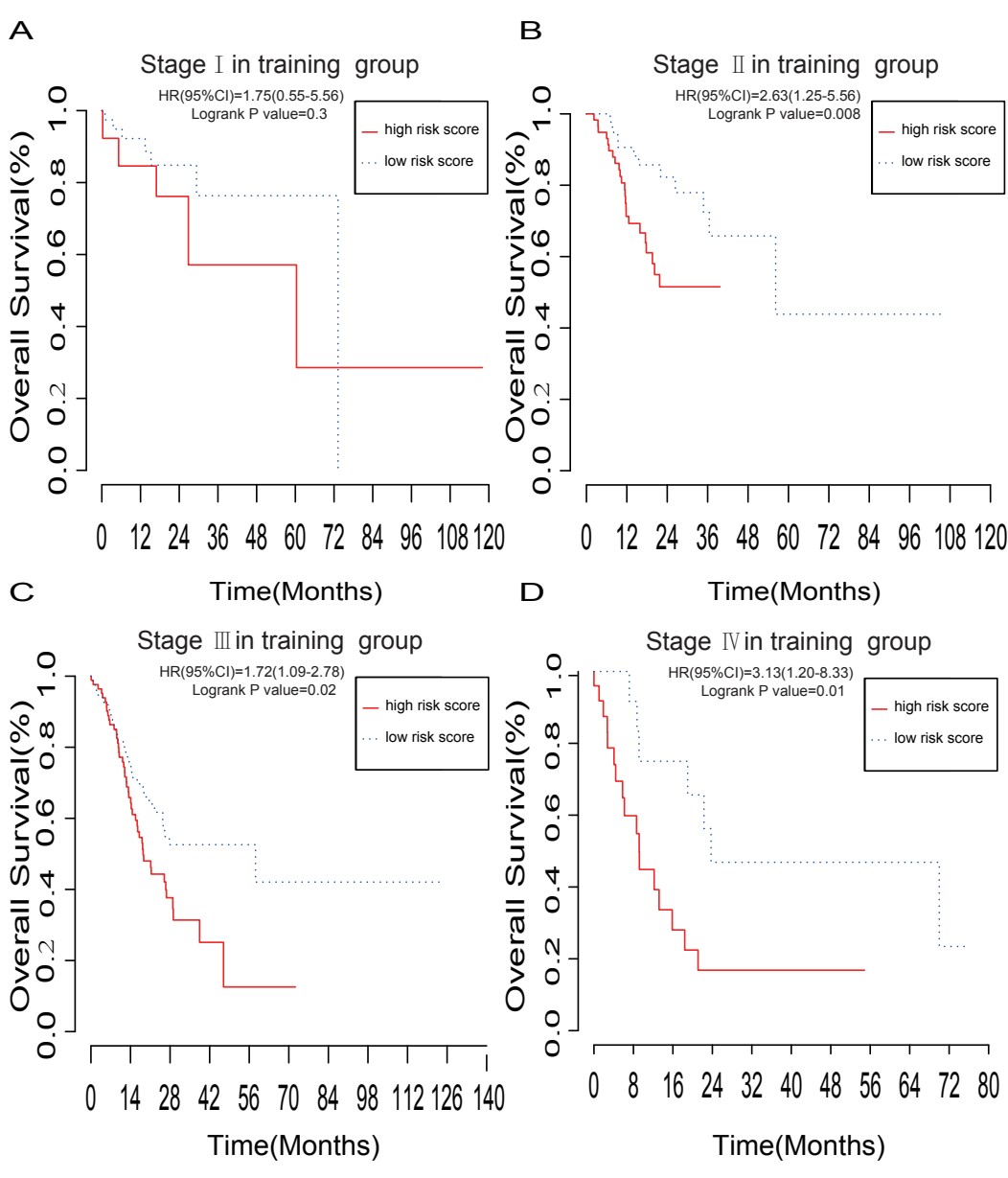

**Figure 4** **The prognostic value of lncRNA-based risk model in subgroups according to the TNM stage.**
(A–D) Kaplan–Meier analysis of the OS of GC patients with stage I, II, III and IV, respectively.

cancers, and can also be used to predict patient prognosis. Several nomogram models involving clinical factors have been constructed to predict the prognosis of patients with GC. For example, Yu (*Yu & Zhang, 2019*) used tumor size and tumor site, as independent prognostic factors, to construct OS nomograms for predicting outcomes in patients with GC, and the C-index of this model indicated that it could predict prognosis. Recently, more lncRNAs related to GC prognosis have been discovered; however, prognostic prediction models involving lncRNAs still lack consensus. We present a nomogram including clinical

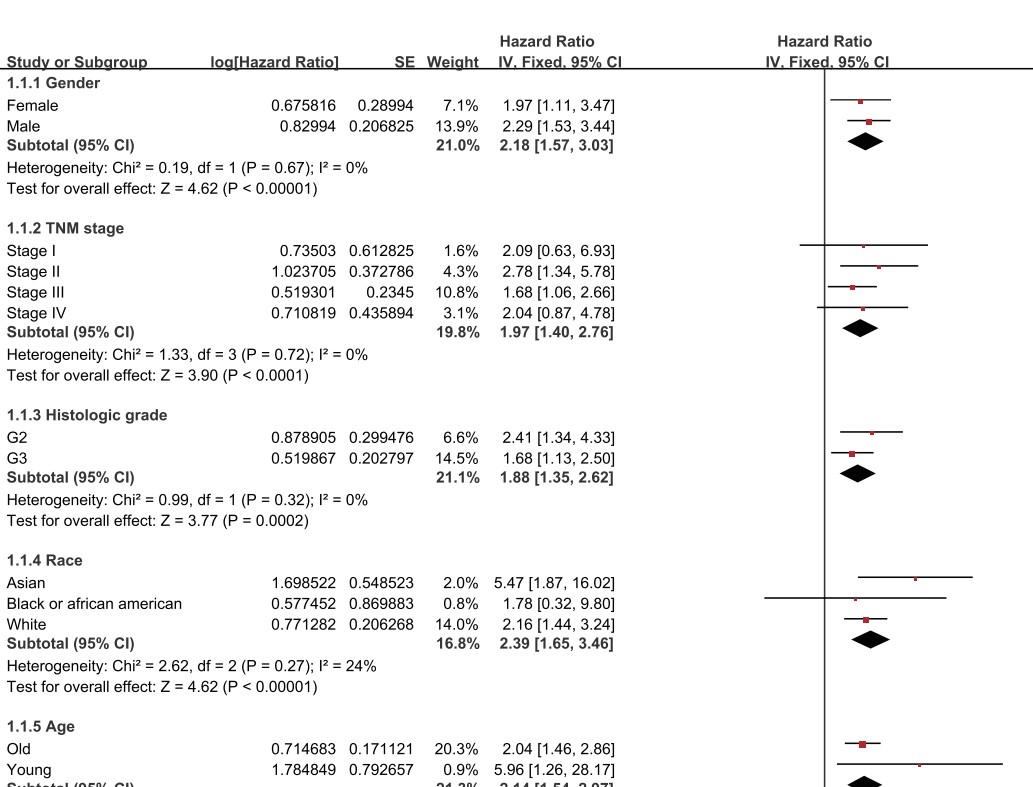

**Figure 5** **Forest plot to evaluate prognostic value of lncRNA-based risk model in subgroups divided by clinical factors.**

factors and the five-lncRNA signature that might be of value for prognostic prediction in GC.

It is necessary to explore novel biomarkers to improve the diagnostic accuracy and prognosis of GC because of limitations of TNM staging and some related scoring systems. Many lncRNAs have been identified, of which only few have been functionally annotated. However, evidence indicates that lncRNAs, acting either as oncogenes or tumor suppressors, participate in the tumorigenesis and development of various cancers by regulating chromatin remodeling, transcription and post-transcriptional modification (*Bartonicek, Maag & Dinger, 2016*; *Iyer et al., 2015*), and therefore might be valuable for cancer diagnosis and prognosis. Some studies have found that GC-related lncRNAs are involved in biological behaviors including the proliferation, migration, invasion, and autophagy of GC cells, thereby affecting the initiation and prognosis of GC (*Mao et al., 2017*; *Wei & Wang, 2017*). For example, the lncRNA MEG3 inhibits the proliferation,

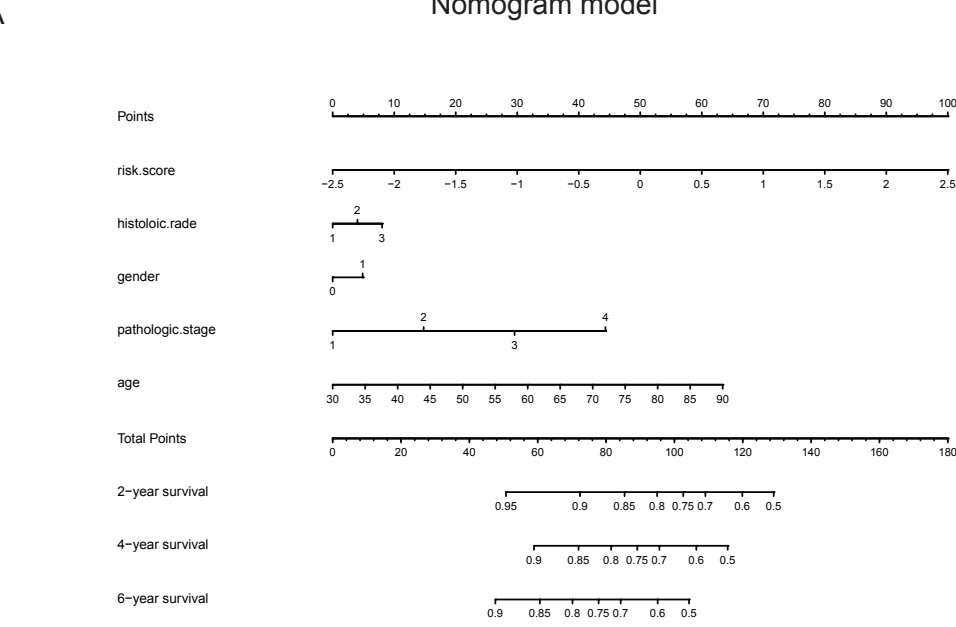

## Nomogram model (A)

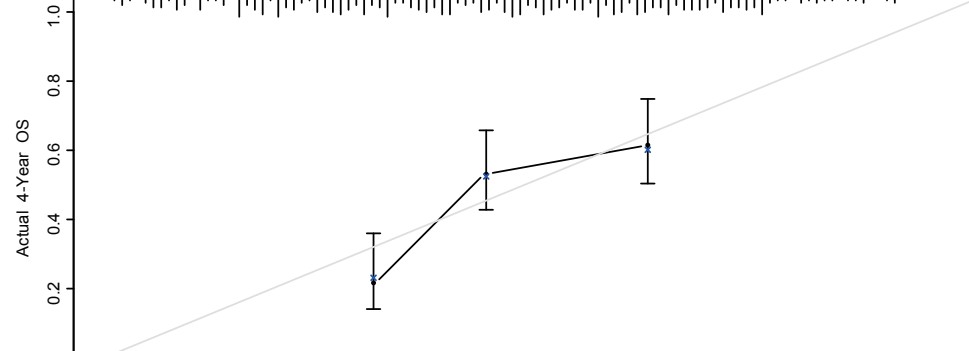

## Nomogram calibration curve (B)

**Figure 6  The prognostic value of a nomogram model combining five-lncRNA signature with the clinical factors.** (A) A nomogram model combining five-lncRNA signature with the clinical factors for predicting the 4-year OS of GC patients. (B) The nomogram calibration curve to evaluate the prediction of 4-year OS of GC patients. The C index of this model was also calculated.

metastasis, and prognosis of GC cells by upregulating the expression p53—a key tumor suppressor (*Wei & Wang, 2017*). We identified five lncRNAs—LINC00205, TRHDE-AS1, OVAAL, LINC00106, and MIR100HG—as predictors of GC prognosis, and developed a risk-score model. Kaplan–Meier analysis suggested that our lncRNA-based risk model is

**Table 4  The association between five-lncRNA signature and OS of GC patients in training group.**

| | Number (High Risk score/Low Risk score) | HR (95% CI) | *P* value |
|---|---|---|---|
| **Total** | 204/204 | 2.09 (1.80, 2.44) | 0.000001 |
| **Gender** | | | |
| Male | 129/134 | 2.29 (1.53, 3.44) | 0.00002 |
| Female | 75/70 | 1.97 (1.11, 3.47) | 0.01 |
| **Histologic grade** | | | |
| G2 | 47/97 | 2.41 (1.34, 4.33) | 0.0006 |
| G3 | 146/97 | 1.68 (1.13, 2.50) | 0.02 |
| **Race** | | | |
| Asian | 44/41 | 5.47 (1.87, 16.02) | 0.001 |
| Black or african american | 4/8 | 1.78 (0.32, 9.80) | 0.6 |
| White | 138/120 | 2.16 (1.44, 3.24) | 0.0003 |
| **Age** | | | |
| Old ($\geq$50 years old) | 186/191 | 2.04 (1.46, 2.86) | 0.00001 |
| Young(<50 years old) | 18/13 | 5.96 (1.26, 28.17) | 0.008 |
| **TNM stage** | | | |
| Stage I | 14/41 | 2.09 (0.63, 6.93) | 0.3 |
| Stage II | 62/58 | 2.78 (1.34, 5.78) | 0.008 |
| Stage III | 88/79 | 1.68 (1.06, 2.66) | 0.02 |
| Stage IV | 25/16 | 2.04 (0.87, 4.78) | 0.01 |

**Notes.**

HR, Hazard ratio; 95%CI, 95% confidence interval.

valuable for predicting GC prognosis. We used two independent GEO datasets as validation datasets. Our results confirmed that our risk score model is stable and performs well in the prognostic prediction of GC.

Of the five lncRNAs, LINC00205, TRHDE-AS1, OVAAL, and MIR100HG, act as risk factors of GC, whereas LINC00106 is a protective factor. Apart from LINC00205 and MIR100HG, the other three lncRNAs have not been reported much in the literature. Our study identified LINC00205, TRHDE-AS1, OVAAL, and MIR100HG as potential prognostic biomarkers of GC for the first time. Consistent with our result, it has previously been reported that high expression of LINC00106 indicates prolonged OS of patients with GC (*Qi et al., 2020*). Nevertheless, the role of this lncRNA in GC as well as its specific mechanism need to be further investigated. Interestingly, in hepatocellular carcinoma (HCC), comprehensive genome-wide analysis revealed that the expression of LINC00205, a tumor suppressor, is positively associated with OS and recurrence-free survival (*Cui et al., 2017*). A study showed that, as a competing endogenous RNA with lower expression levels in tumor tissues, LINC00205 may negatively regulate HCC progression via the miR-184/EPHX1 axis (*Long et al., 2019*), While another study indicated that LINC00205, can serve as an oncogene, and can promote the proliferation, migration and invasion of HCC cells by targeting miR-122-5p (*Zhang et al., 2019a*). In addition, LINC00205 can act as a protective factor in pancreatic cancer [HR = 0.58, *P* (log rank) = 0.0091] (*Giulietti et*

**A**

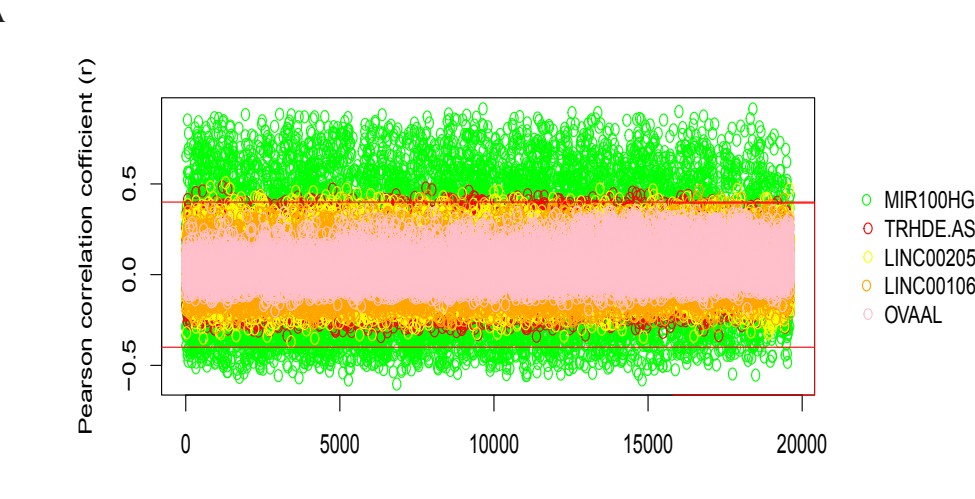

**B**

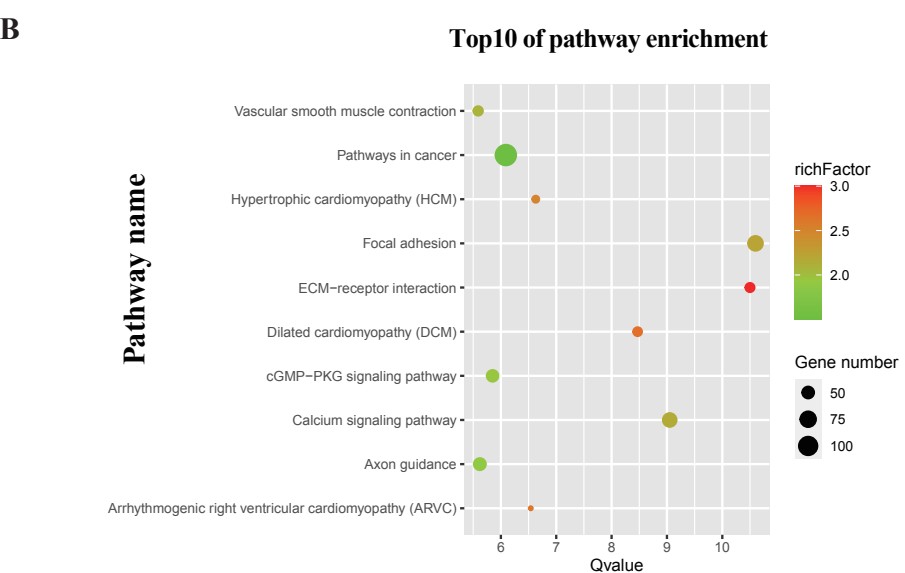

**Figure 7  Potential functions of the five lncRNAs.** (A) The Pearson correlation coefficient between 19,605 protein-coding genes and five lncRNAs in TCGA dataset. (B) The functional enrichment bubble map of pathways by KEGG pathway analysis. Bubble size represents the number of gene enriched in the pathway.

*al., 2018*). The reported role and therefore prognostic prediction value of LINC00205 in various cancers shows significant discrepancies. These discrepancies might be associated with the specificities of different cancers. The upregulation of TRHDE-AS1 inhibits the growth of lung carcinoma through competitive combination with the miRNA-103-KLF4 axis (*Zhuan et al., 2019*). A study has found that OVVAL is highly expressed in colon cancer and melanoma, and further experimental results showed that OVAAL promotes the proliferation of cancer cells via dual mechanisms controlling RAF/MEK/ERK signaling

and p27-mediated cell senescence (*Sang et al., 2018*). The lncRNA MIR100HG has been studied as an oncogene in acute megakaryoblastic leukemia (*Emmrich et al., 2014*), and laryngeal squamous cell carcinoma (*Huang, Zhang & Zhou, 2019*), as well as for its role in mediating cetuximab resistance via Wnt/ β-catenin signaling (*Lu et al., 2017*) in colorectal cancer. Although the roles of these lncRNAs in cancer need to be further investigated, our results may provide a novel approach to study GC.

To further investigate the functions of the five lncRNAs in GC, we performed pathway enrichment analysis. These genes are enriched in cancer regulation, including the cGMP−PKG signaling pathway, calcium signaling pathway, and focal adhesion pathway etc. This finding suggests that the five lncRNAs may play an important role in the occurrence and development of GC. There is evidence that lncRNAs can promote tumorigenesis through the cGMP−PKG signaling pathway. For example, the overexpression of SRRM2-AS accelerates angiogenesis in nasopharyngeal carcinoma via the cGMP−PKG signaling pathway (*Chen et al., 2019*). The calcium signaling pathway has been reported to be mainly involved in metabolic diseases and heart diseases over the past years (*Berridge, 2016*; *Dewenter et al., 2017*). A recent research showed that the calcium signaling pathway was associated with cancer cell survival, but more details on its effects remain to be studied (*Reczek & Chandel, 2018*). Focal adhesion sites are special sites where integrin receptors aggregated in cells interact with extracellular matrix and intracellular actin skeleton (*Burridge, 2017*), and they play a critical role in tumor invasion and migration (*Shen et al., 2018*). There is evidence that knockdown of Linc01060 could promote the progression of pancreatic cancer via the vinculin-mediated focal adhesion pathway turnover (*Shi et al., 2018*). However, whether the lncRNA can mediate the progression of GC through the focal adhesion pathway is less reported. In short, lncRNAs may participate in the genesis and development of various tumors via the above pathways.

Risk score model is a common and widely used method to predict the prognosis of patients with multiple diseases (*Lemke et al., 2017*; *Li et al., 2018*; *Yang et al., 2017*; *Sobotka et al., 2018*). Our risk score was determined by the expression of independent survival-lncRNAs obtained after Cox hazard analyses and its corresponding coefficients. It was calculated using binary lncRNA expression values according to the medians of original lncRNA expression values. This adjustment helps to improve the clinical application of the prognostic model in other study population (*Zhang et al., 2018*). In general, the higher is the risk score, the poorer is the prognosis, which is consistent with our analysis. Our Kaplan–Meier survival analysis showed that the patients in the high risk group had a significantly poorer prognosis than those in the low risk group. Our risk score model based on lncRNAs has several advantages. This model based on the expression of five lncRNAs provides a novel noninvasive method for predicting the prognosis of patients with GC before surgery. Compared with conventional invasive pathological examinations, it reduces unnecessary pain for patients. Second, this five-lncRNA risk model can provide preoperative risk predictive probability of individual mortality and recurrence in different clinical endpoints. It is simple and convenient for clinicians and patients to understand. Third, our model used the median of five-lncRNA-based risk score as the cutoff value to divide patients into high risk and low risk groups. It can identify patients at high risk of

mortality or recurrence in a timely manner and prompt clinical interventions as early as possible to improve their prognosis.

There have been several reports on lncRNA signatures for GC. A previous study reported a 24-lncRNA signature that can predict outcomes in patients with GC by applying the random survival forest-variable hunting algorithm using GEO datasets (*Zhu et al., 2016*). However, because of the limited amount of data in the GEO datasets, the lncRNAs identified in this study might not represent the complete population of lncRNAs involved in GC. In this study, we integrated 950 samples from TCGA and GEO databases to comprehensively investigate the potentially prognostic lncRNAs. This greatly improved the accuracy, reliability and robustness of our model. A six-lncRNA prognostic signature was established by robust likelihood-based survival and LASSO model (*Ma, Li & Ren, 2019*). Whether the six-lncRNA signature combined with other clinical features can enhance the predictive power remains to be determined. To improve the accuracy of the five-lncRNA prognosis model, we combined it with clinical factors to develop a nomogram model that could predict the OS of patients with GC. *Zhu et al. (2018)* et al. constructed an 11-lncRNA signature by univariate and multivariate Cox regression analyses. Although an internal validation was validated using the bootstrap resampling method, external validation studies are needed to further evaluate the value of this model. We not only included two external verification datasets, but also performed survival analysis, ROC curve analysis, and constructed a forest plot for predictive verification, indicating a favorable effectiveness of our model.

There are some limitations of the present study. We integrated data from TCGA and GEO databases to increase the number of the cases, thereby reducing bias from a small sample size. Integrated analysis has been proved to be an effective approach for multiple datasets with different platforms using R package (*Zhang et al., 2019b*; *Nie et al., 2020*; *Zhao et al., 2018*; *Zhao et al., 2020*), thereby promoting the reliability of our conclusion (*Ma et al., 2017*). However, TCGA dataset has a larger number of lncRNAs than the GEO dataset (14147:1397) because of different sequencing technologies: TCGA uses RNA sequencing technology, whereas GEO uses microarray chip technology. Intersection of three datasets has inevitably omitted potential prognostic lncRNAs. Moreover, the clinical characteristics of the patients in the three datasets are heterogeneous. This might have inevitably led to a bias. Besides, owing to the lack of DFS and clinical data in GSE15459, we used only one external validation group to verify the prognostic value of the five-lncRNA signature for the DFS of patients. In addition, many important variables affecting the prognosis of patients with GC are not provided in TCGA and GEO datasets, such as dietary habits, previous disease, history of chemotherapy or radiation therapy, and family history of cancer. Thus, on the one hand, it is necessary to perform a large-scale multi-center prospective clinical study based on the same sequencing technology to decrease the bias mentioned above. On the other hand, based on existing data, it is beneficial to develop innovative statistical algorithms to reduce the heterogeneity of different data sources. Last, because of the limited number of studies regarding these lncRNAs, experimental research on these lncRNAs is highly warranted to further understand their functions in GC.

## CONCLUSIONS

We established a risk score model including five lncRNAs to predict the OS and DFS of patients with GC, particularly in those with stage II-IV GC. Our findings also provided evidence of developing effective prognostic biomarkers for patients with GC and potential therapeutic targets in the future.

## ACKNOWLEDGEMENTS

At the point of finishing this paper, we would thank to Jie Peng for technical assistance with the data analysis.

### Funding

This work was supported in part by grants from the Science and Technology project of Health Commission of Jiangxi Province, China (202130409) and the Science and Technology project of Jiangxi Provincial Administration of Traditional Chinese Medicine, China (2020A0051). The funders had no role in study design, data collection and analysis, decision to publish, or preparation of the manuscript.

### Grant Disclosures

The following grant information was disclosed by the authors:
Science and Technology project of Health Commission of Jiangxi Province, China: 202130409.
Science and Technology project of Jiangxi Provincial Administration of Traditional Chinese Medicine, China: 2020A0051.

### Competing Interests

The authors declare there are no competing interests.

### Author Contributions

- Yiguo Wu conceived and designed the experiments, performed the experiments, analyzed the data, prepared figures and/or tables, authored or reviewed drafts of the paper, and approved the final draft.
- Junping Deng and Shuhui Lai performed the experiments, analyzed the data, prepared figures and/or tables, and approved the final draft.
- Yujuan You conceived and designed the experiments, authored or reviewed drafts of the paper, and approved the final draft.
- Jing Wu conceived and designed the experiments, performed the experiments, authored or reviewed drafts of the paper, and approved the final draft.

### Data Availability

Data are available at UCSC Xena (https://xenabrowser.net/) and at NCBI GEO: GSE62254 and GSE15459.

## Supplemental Information

Supplemental information for this article can be found online at http://dx.doi.org/10.7717/peerj.10556#supplemental-information.

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
