# Peer review of "A risk score model with five long non-coding RNAs for predicting prognosis in gastric cancer: an integrated analysis combining TCGA and GEO datasets"

_PeerJ, doi:10.7717/peerj.10556_

## Round 0.1 · original submission · Major Revisions

Both reviewers believe the data reported and methods you have used are sound, and I believe your manuscript would be of interest for publication by PeerJ subject to addressing the comments from both reviewers. In revising your manuscript, I would like to ask you to provide a point-by-point response to the reviewer comments with a separate document indicating where changes were made (e.g. using highlighting or tracked-changes).

As you can also see from comments from both reviewers (their comments included are below), there is a need to thoroughly proof-read the manuscript and correct type-o's and address grammatical and spelling errors. As PeerJ does not offer copy editing, therefore I ask if you could do this, perhaps by asking local colleagues to proof read the manuscript to make sure the use of English is correct.

Both reviewers have identified instances where figures or the text does not have sufficient detail, therefore I encourage you to address all of these points and make the appropriate changes to the text, figures and tables. At the bare minimum comments 4, 6, 7, 8, 11 and to expand the limitations from reviewer 1 and comments Basic 1, 3, 4; Experimental 1-4; and Validity 1-3, 5 from reviewer 2 must be appropriately addressed in a revision, with all remaining comments discussed in a rebuttal.

I look forward to receiving a revised manuscript in due course (and wish you all the best of health during these times).

Reviewer 1 ·

Basic reporting

The authors' attempt and purpose to build a risk prediction model in the context of Gastric Cancer seems logical and clear as such type of models is not uncommon in biomedical research. They seek to aid clinicians and patients to undertake better decisions about GC
treatments.

Experimental design

All data sources, statistical packages, and computational models are explicitly named in the paper and available for the study replication. Overall, the data and methods parts are adequately presented and easy to follow. The research is multistep describing each part in sufficient detail across the article. The statistical methods include Bayesian analysis, Cox regression and Kaplan-Meier all of which are frequently applied in biomedicine and seem appropriate for performing their respective tasks at different stages of the project to create a scoring model and identifying important gastric cancer gene-related biomarkers. Every R package used in the study is specified, making it easier for other researchers to reproduce the study. Even though the methodology and data are clearly described in the article, the whole text will benefit from proofreading to eliminate typos and various
grammatical issues that appear across various sections including Figures. Nevertheless some presentations of methods I found a bit difficult to grasp, despite knowing something about their practical application.

Validity of the findings

1. More information should be provided on why the authors bounded the OS and DFS at 48 and 24 months respectively. The description: “the curve tends to be gentle” (line 138) is subjective. There is still at least a 10% decrease in both survival curves after those bounds (Figure 1 C-D).

2. Survival curves (figure 1-4) should include a confidence interval. This is specifically important given that the size of the datasets is small.

3. It is not clear how the authors concluded that the small sample size in stage (1) patients is a limitation (line188) without carrying out a power analysis.

4. Table 2: showing a representation of the workflow; somehow lacks the univariate cox proportional hazard regression analysis which is
described (line 129).

5. Table 2: layout should be modified as the directions of arrows follows a snake-like structure making it difficult to read. In addition, each rectangular box could be numbered to improve reading simplicity.

6. Table 3: number of patients doesn’t match the text i.e. (line 189) says that a number of patients in stage I is 26, yet table 3 shows that stage I has 14 high and 41 low-risk patients. Please clarify

7. Age cut off for Old / Young is not specified (would be good to have
clarification).

8. P-value was adjusted (good to have clarification on how they were adjusted).

9. Figure 7: how the top 10 enrichment pathways were selected?

10. It might be more informative to include the clinical information
for each cancer stage separately in table 3.

11. Could be explained how the Kyoto Encyclopaedia of Genes and Genomes was combined with the Web-based Gene set analysis toolkit to analyse the aberrantly activated signalling pathways.

Also important to mention that the deficiencies in the statistical applications are only succinctly mentioned by the authors. In my view, it could have been identified in the article more meaningfully. Expanding it specifically to statistical methods as well as broadly
discussing the shortcomings of data would make the paper more valuable.

Reviewer 2 ·

Basic reporting

1.LncRNAs and gene mentioned in the materials, methods and results section of the paper may express the same meaning. It is suggested to explain clearly and avoid confusion.
2.Please provide the lists of 278 lncRNAs and 38 shared genes in the supplementary materials.
3.Please correct the spelling mistakes such as time-dependent and forest plot.
4.Figure 1B is not clear and please add the legend to explain it.

Experimental design

1.Please confirm the the cut-off value of high and low risk groups is the median or zero score in Figure 2C-D and 3C-D further.
2.How are 408 samples screened from the TCGA STAD database?
3.DFS data of GSE62254 dataset is available from the supplementary materials of the article (Molecular analysis of gastric cancer identifies subtypes associated with distinct clinical outcomes), why is it not verified?
4.Please provide the details of the risk score model and the risk score of each patient to explain the range of risk score (-2.5-2.5) in the nomogram model.

Validity of the findings

1.Please provide the multivariate cox analysis form of TCGA data.
2.The coefficients of 5 lncRNAs do not correspond to the HR values in Table 1.
3.The sample sizes of the scatter plot (n = 300 and 140) are not consistent with that of data sets (n = 300 and 200) in the validation groups.
4.It is recommended to give the ROC analysis of 1-4 years cut-off OS and 1-2 years cut-off DFS.
5.In Figure 7A, the absolute value of Pearson correlation coefficient should not be greater than 1.

Additional comments

The article showed a risk score model combining five-lncRNA signature to predict the prognosis of GC patients, while there were still some work remaining to be improved.

---

## Round 0.2 · Minor Revisions

Please ensure you respond to each comment made by the two reviewers and indicate how you have made the clarifications in the manuscript text. This includes typos and consistencies as indicated by reviewer 1 as PeerJ does not perform proofing as standard. Reviewer 2 makes a good point and I recommend you explain in the discussion how the risk score should be interpreted going forward.

In order to allow others to evaluate your work, it is beneficial to provide sufficient details to allow replication. This includes indicating what versions of software packages were used, whether results relate to training or test sets, as well as to detail metadata of the study cohort.

Reviewer 1 ·

Basic reporting

1. There are still inconsistencies in the paper that need to be addressed. For instance in lines 175-176 “OS and DFS in test cohort (Fig. 2A-B)”, but figure 2 description “(A-B)
Kaplan-Meier analysis of patients’ overall survival and disease-free survival in the high-risk (n = 204) and low-risk (n = 204) subgroups of the training set.” It should be clarified to whether the graphs and figures are obtained from the training or test dataset.

2. Even though the paper has been proofread, further work is still required. For eaxample: line 243 “differented” and line 258 “socring”. There are also inconsistencies with the words used: line 182-184 author uses “cut-off”, but in lines 108 and 116 “cutoff” is used instead. Another example is with the word dataset: line 340 “datasets” but in line 341 author uses “data sets”. There are similar mistakes within the text that should be carefully reviewed.

Experimental design

'no comment'

Validity of the findings

The validity of finds is appropriately, and it's evaluated on an
external dataset (GEO).
1. The paper would benefit if it included metadata on all patients, not only the ones in the training cohort, as seen in Table 3.

2. The discussion section should consist of more information regarding the limitations of the study/datasets and how it would be possible to improve them in the future.

Reviewer 2 ·

Basic reporting

No comment.

Experimental design

Please confirm the the cut-off value of high and low risk groups in Figure 3C-D further.

Validity of the findings

According to the risk score model, how to understand the risk score ranges from -2.086959745 to 2.270305234?
Whether LINC00106 plays an important role in the model?

Additional comments

The article showed a risk score model combining five-lncRNA signature to predict the prognosis of GC patients, while there were still some work remaining to be improved.

---

## Round 0.3 · Minor Revisions

I believe most comments from reviewers have been addressed.
However there are still changes to be made with regards to proofing and to improve readability/reproducibility:

E.g. in the title 'combing' should be 'combining' (and 'dataset' should be 'datasets' (plural)). In the methods (line 88) 'recruited' should be 'included', the datasets are publicly available and recruitment has already happened (this is correct in the results, line 149).
In the abstract and methods the authors talk about using 'Bayesian analysis' - what type of Bayesian analysis/model is this (empirical Bayes?) and can they detail how the P-value (a frequentist not a Bayesian concept) is obtained from the Bayesian analysis? Simply making reference to 'Bayesian analysis' is too broad and does not allow replication.

Considering most of the references in the introduction relate to Chinese GC studies, and the GEO datasets are from Singaporean and East Asian cohorts, how does TCGA data from predominantly white US Americans reflect this, can the authors add an authoritative reference to ensure the results and translatable?

---

## Round 0.4 · accepted · Accept

Table 3 (The schematic workflow of the present study) still includes 'Bayesian Analysis' in the methods used (step 2), please could you change this to reflect how you've changed the methods as detailed in your rebuttal?